# Muramyl Dipeptide Administration Delays Alzheimer’s Disease Physiopathology via NOD2 Receptors

**DOI:** 10.3390/cells11142241

**Published:** 2022-07-19

**Authors:** Pierre-Alexandre Piec, Vincent Pons, Paul Préfontaine, Serge Rivest

**Affiliations:** Neuroscience Laboratory, CHU de Québec Research Center, Department of Molecular Medicine, Faculty of Medicine, Laval University, 2705 Laurier Boul., Québec, QC G1V 4G2, Canada; pierre-alexandre.piec@crchudequebec.ulaval.ca (P.-A.P.); vincent.pons@crchudequebec.ulaval.ca (V.P.); paul.prefontaine@crchudequebec.ulaval.ca (P.P.)

**Keywords:** NOD2, MDP, Alzheimer’s, microglia, monocytes, innate immunity, amyloid, phagocytosis, cognitive decline, immunomodulation

## Abstract

Alzheimer’s disease (AD) is the most common form of dementia in the world. The prevalence is steadily increasing due to an aging population and the lack of effective treatments. However, modulation of innate immune cells is a new therapeutic avenue, which is quite effective at delaying disease onset and improving cognitive decline. Methods: We studied the effect of the NOD2 receptor ligand muramyl dipeptide (MDP) on the modulation of the innate immune cells, namely patrolling monocytes and microglia. We administrated MDP once a week for 3 months in an APP_swe/PS1_ mouse model in both sexes. We started the treatment at 3 months before plaque formation and evaluated its effects at 6 months. Results: We showed that the MDP injections delay cognitive decline in both sexes via different mechanisms and protect the blood brain barrier (BBB). In males, MDP triggers the sink effect from the BBB, leading to a diminution in the amyloid load in the brain. This phenomenon is underlined by the increased expression of phagocytosis markers such as TREM2, CD68, and LAMP2 and a higher expression of ABCB1 and LRP1 at the BBB level. The beneficial effect seems more restricted to the brain in females treated with MDP, where microglia surround amyloid plaques and prevent the spreading of amyloid peptides. This phenomenon is also associated with an increase in TREM2 expression. Interestingly, both treated groups showed an increase in Arg-1 expression compared to controls, suggesting that MDP modulates the inflammatory response. Conclusion: These results indicate that stimulation of the NOD2 receptor in innate immune cells is a promising therapeutic avenue with potential different mechanisms between males and females.

## 1. Introduction

Alzheimer’s disease (AD) is the most common neurodegenerative disease and the most common form of dementia in the world [1]. The number of cases is constantly growing due to a lack of reliable diagnosis and effective treatment [2]. The disease is multifactorial, meaning that it is caused by several factors such as the environment, genetics and epigenetics, and pre-existing conditions [3]. One of the characteristics of AD is the deposition of beta-amyloid (Aβ) in the brain and the vasculature [4]. This small peptide derives from the cleavage of the amyloid precursor protein and can be found in two main forms, namely Aβ_40_ and Aβ_42_ [5]. Aβ_42_ is the preferential form in senile plaques, whereas Aβ_40_ primarily accumulates in the brain blood vessels, which leads to dysregulated neuroinflammation and to the leakage of the brain–blood barrier (BBB), micro-hemorrhages, and small cerebrovascular accidents [6,7].

Innate immunity was suggested to play a role in AD and participates directly in the progression of the pathology. Indeed, amyloid deposits activate microglia, the immune cells of the central nervous system. These activated microglia synthesize inflammatory factors such as tumor necrosis factor-alpha, interleukin-1beta, and reactive oxygen species, which have been suggested to harm neurons [8,9]. On the other hand, monocytes are also in contact with amyloid in blood vessels and participate in the inflammatory response, leading to the BBB breakdown and the infiltration of monocytes in the brain [10]. 

In the last decade, researchers have shown great interest in the role of innate immunity in AD. Microglia and monocytes are overwhelmed by the accumulation of amyloid levels during the disease progression and are enrolled in a vicious circle of inflammation [11,12]. The idea was then to use molecules able to regulate the immune system, such as MPL and mCSF. Both provided impressive results by delaying the cognitive decline associated with AD [13]. These studies demonstrated a powerful effect of delaying AD onset by modulating innate immune cells. These studies also showed that stimulation of a specific monocyte subtype, namely patrolling monocytes/non-classical monocytes, is a promising therapeutic avenue [14,15,16].

Patrolling monocytes arise from bone marrow and are characterized by the expression of CX3CR1, NOD2, and CSF1R and have a greater ability to phagocyte and regulate inflammation than other cell types [14,17,18]. Additionally, these monocytes are CCR2^−^ and Ly6C^−^ and recent studies using intravital microscopy showed that patrolling monocytes crawl on the wall of blood vessels and can phagocyte amyloid along the luminal side of the BBB [15]. Lately, Lessard and colleagues demonstrated that muramyl dipeptide (MDP), when injected in mice, can drastically increase the number of patrolling monocytes in the blood by converting classical monocytes to non-classical monocytes [18]. This switch is mediated by the nucleotide-binding oligomerization domain-containing 2 (NOD2), but the exact mechanism remains unclear. However, some articles proposed that MDP-stimulated NOD2 converts monocytes in an Nur77-dependent manner [2,18,19]. Interestingly, mice lacking either Nur77 or NOD2 showed a defect in the production and conversion of patrolling monocytes, suggesting that both receptors are required for the switch and survival of non-classical monocytes [20].

Only one study so far has evaluated the role of MDP in an AD mouse model. Maleki and colleagues demonstrated that modulation of the immune system with MDP has a significant potential to delay AD in male mice [21]. 

In this study, we hypothesized that MDP-stimulated monocytes and microglia can delay the cognitive decline in AD in a sex-dependent manner. We used the well-known strain APP_swe/PS1_, later named the APP mouse. We primed the immune system with one injection of MDP per day for three consecutive days, and then once a week. We aimed to evaluate the effect of the molecule on AD onset from 3- to 6-month-old animals (Figure 1). At 6 months, mice have a poor short-term memory associated with a tremendous amount of amyloid within the brain and some amyloid deposit in blood vessels. We found that MDP injections delayed the short-term memory loss in treated APP mice at 6 months of age compared to their control littermates in both sexes, which was associated with a strong protection of the BBB in all MDP-treated groups. Interesting, we found differences between both sexes, especially when quantifying the amyloid load in the cortex and hippocampus and the expression of phagocytosis markers. 

Altogether, these results show that MDP has a beneficial effect on cognitive decline and BBB protection and NOD2 ligands could be considered as a promising therapeutic avenue for both males and females suffering from AD. 

## 2. Materials and Methods

### 2.1. Animals

WT mice on a C57BL/6J background were purchased from The Jackson Laboratory (Bar harbor, ME, USA). For this study, we used male and female APP_swe/PS1_ mice. These were transgenic mice bearing a chimeric human/mouse β-amyloid precursor protein (APPSwe) gene and the human presenilin 1 gene PS1. These mice were purchased from The Jackson Laboratory [Strain: B6C3-Tg (APP695)3Dbo Tg(PSEN1)5Dbo/J] and maintained on a C57BL/6J background. Mice were injected once a day for 3 consecutive days with either MDP (Invivo Gen San Diego, CA, USA, #tlrl-MDP) diluted in saline (10 mg/kg) or vehicle (saline 0.9%) intraperitoneally and then once a week for 3 months. All protocols were performed according to the Canadian Council on Animal Care guidelines, administered by the Laval University Animal Welfare Committee (2020-402, CHU-20-402). NOD2 mice, B6.129S1-*Nod2^tm1Flv^*/J were purchased from The Jackson Laboratory. WT and NOD2^−/−^ mice were used to assess the role of NOD2 in the switch of monocytes (*n* = 4/group, 2 groups only males), whereas APP and WT mice were used to assess the role of MDP in AD (*n* = 8/group, 6 groups, 3 female groups and 3 male groups).

### 2.2. Sacrifices

All mice were deeply anesthetized with ketamine/xylazine (90:10) and sacrificed via intracardiac perfusion with PBS 1X. Brains were retrieved and postfixed for 24 h at 4 °C in 4% PFA pH 7.4 and then transferred in 4% PFA pH 7.4 +20% sucrose for a minimum of 15 h. Brains were sliced in coronal sections with a 25-μm thickness with a freezing microtome (Leica Microsystems, Deerfield, IL, USA), serially collected in anti-freeze solution, and kept at −20 °C until usage.

### 2.3. Western Blot

Brain protein lysate was extracted and quantified as previously described [22]. Proteins were loaded in 8–16% agarose precast gels (BioRad, Montreal, Qc, Canada) and electroblotted onto 0.45-µm Immibilon PVDF membranes. Membranes were incubated with primary antibody at a concentration of 1/1000 overnight at 4 °C followed by the appropriate horseradish peroxidase (HRP)-conjugated secondary antibodies and revealed by clarity (ECL) substrate (BioRad) (Table 1). Quantification was carried out by determining the integrative density of bands using ImageJ software. Optical values were normalized over actin. 

### 2.4. Flow Cytometry

Flow cytometry is a method that is used to examine and determine the expression of specific receptors on the cell surface. This allows the definition and characterization of distinct single cell types such as monocytes, lymphocytes, and neutrophils depending on the expression of specific markers. Flow cytometry analysis was performed on PBMC cells for analysis. Blood samples were collected from the submandibular vein with an 18 G needle and kept in EDTA (SARSTEDT Saint-Louis, MI, USA#20.1341.100)-coated vials on a rotator for <1 h. In total, 60 µL of the blood samples was transferred to a 5 mL Falcon (Falcon Cambridge, MA, USA #352054) with 1 mL of ACK lysing Buffer (Gibco Cambridge, MA, USA #A10492-01) for 20 min. This was followed by washing with PBS (Multicell Cambridge, MA, USA #311-010-CL) and centrifugation for 10 min at 4 °C and 500 G. After discarding the supernatant, we blocked non-specific sites with 200 µL of PBS containing 2 µL of CD16/CD32 (BD^TM^ Bioscience Toronto, On, Canada #553142) on ice for 10 min.

This was followed by the addition of 100 µL of PBS containing 1µL of each fluorochrome panel on ice in the dark for 40 min. Compensation tubes were prepared in parallel with 1µL of one antibody with 1 drop of Arc reactive Beads (Invitrogen Waltham, MA, USA #A10346) for the live/dead antibody, with one drop of Ultracomp eBeads (Invitrogen #01-2222-42). We then washed the samples by adding 3 mL of PBS to the samples and the live/dead compensation tube and theses tubes were centrifuged at 500 G and 4 °C for 10 min. We discarded the supernatant and added 300 µL of PBS and 50 µL of 123count eBeads™(Invitrogen#01-1234-42). We added 300 µL of PBS to the compensation tubes for the live/dead and 1 drop of Arc negative beads (BD^TM^ CompBeads Toronto, ON, Canada#51-90-9001291). The samples were then ready to be analyzed.

The monocyte subsets were selected by a combination of Blue-fluorescent reactive Dye (Invitrogen, #L23105A), Ly6G-PE (BD Horizon^TM^ #551461, (BLy6C-V450 (BD Horizon^TM^ #560594), CD45-V500 (BD Horizon^TM^ #561487), and CD11b-AF700 (Invitrogen, #56-0112-82) (Figure 2). FACS and data acquisition were performed using SORP LSR II and FACSDiva software (Francklin Lake, NJ, USA) (both from BD). The gating strategy is described in the Appendix A. The results were analyzed with FlowJo softwareAshland, OR, USA (v10.5.3).

### 2.5. Behavioral Analysis

The novel object recognition task was previously described in [23]. The experimenter who observed and recorded the behavior was not aware of the treatment and genotype of the tested animals. The mouse movement was captured with ANY-maze (Soetling, CO, USA) (version 7) in the gray box, APP-vehicle *n* = 8; and APP-MDP *n* = 8, WT *n* = 8. Baseline data were obtained at 3 months old, whereas the test was carried out at 6 months. 

### 2.6. Immunofluorescence Staining

Brain sections were washed (4 × 5 min) in KPBS then blocked in KPBS containing 1% BSA and 1% Triton X-100 (Sigma-Aldrich, Cambridge, MA, Boston cat#T8787) for 1 h at room temperature. After washing sections in KPBS (4 × 5 min), tissues were incubated with the appropriate primary antibody 1/1000 (6E10, Iba-1) overnight at 4 °C. The next day, after 3 washes with KPBS, tissues were incubated with the appropriate secondary antibodies for 2 h at room temperature. Following by washes in KPBS (4 × 5 min), tissues were incubated for 10 min with DAPI (Molecular Probes Montreal, Qc, Canada 1:10,000 cat#D3571). After washes in KPBS (4 × 5 min), sections were mounted onto MicroSlides Superforst^®^ (Fisherbrand, Watham, MA, USA cat#22-037-246) and cover slipped (Globe scientific, Mawha, NJ, USA cat#1419-10) with Fluoromount-G (Electron microscopy sciences, Cambridge, MA, Boston cat#17984-25).

### 2.7. Unbiased Serological Count

Brains were serially sectioned as previously described and were stained with Iba1, 6E10, and DAPI as previously described. We counted the microglia and plaques at ×20 magnification using an Axio Observer microscope equipped with an Axiocam 503 monochrome (Carl Zeiss Canada, Toronto, ON, Canada) with StereologerTM2000 MFC V.11.0. The sampling characteristics used for the cortex sections were as follows: sampling interval 1, total number of sections 144, section sampling interval 48, and starting selection 1; for the hippocampal sections: sampling interval 1, total number of sections 144, section sampling interval 48, and starting selection 1. After counting plaques, the mean plaque volume (MPV) was estimated using the rotator method. The estimated MPV was based on the length of the line crossing each plaque using a randomly oriented line. Plaques and microglia were counted according to the Gundersen unbiased counting rules using the optical fractionator method and sampling continued to a coefficient of error of 10% or less [23].

### 2.8. HEK-Blue™-hNOD2 Cells

The HEK-Blue™-hNOD2 cells that stably co-express human NOD2 along with the NF-κB SEAP reporter gene were purchased from InvivoGen (San Diego, CA, USA) (#hkd-hnod2). HEK-Blue™-hNOD2 cells are designed to detect the activity of NOD2. The levels of SEAP were measured with HEK-Blue™ Detection (InvivoGen). The cells were cultured and maintained according to the manufacturer’s recommendations. To evaluate NOD2 activation in response to MDP, gradient concentrations of MDP were added to a 96-well plate with 85,000 cells/mL hNOD2 at a concentration of 1 µg/mL. Results are expressed on log[agonist]. NF-κB-induced SEAP activity was assessed in the culture supernatant using HEK-Blue™ and read at OD 650 nm after 6 h. 

### 2.9. CAA Frequency

CAA was quantified using CD31 and 6E10 staining. The number of amyloid-positive vessels in the cortex was manually counted over the 12 whole-brain sections. The count was carried out under blinded conditions regarding the experimental groups. The CAA frequency refers to the number of 6E10-positive vessels [23].

### 2.10. Image Acquisition

Image acquisition of fluorescent staining was performed using a Zeiss LSM800 confocal microscope supported by the Zen software (2.6 systems) using 10, 20, and 40× lenses. Confocal images were then processed using Fiji (ImageJ Version V1.53k). All panels were assembled using Adobe Illustrator San Jose, CA, USA CC 2021.

### 2.11. Statistical Analyses and Figure Preparation

Data are presented as the mean ± standard error of the mean (SEM). Statistical analyses were carried with the Prism software (Version 9.2.0 (283), GraphPad Software Inc. San Diego, CA, USA). Values were considered statistically significant if *p* < 0.05. All data were tested for normality using the d’Agostino-Pearson test. All panels were assembled using Adobe Illustrator CC 2021 San Jose, CA, USA.

## 3. Results

### 3.1. MDP Triggers the Monocyte Conversion in an NOD2-Dependent Manner

We studied the action of MDP on monocytes to evaluate the conversion of inflammatory monocytes to patrolling monocytes. We injected WT mice (*n* = 4) at 3 months of age with MDP (10 mg/kg) once a day for 3 consecutive days. We collected the blood and analyzed the samples using flow cytometry with CD11b, CD45, Ly6C, and live/dead as described in the Appendix A before and after injections. The markers CD11b and CD45 were used to differentiate monocytes from other immune cells. We used the signal strength from Ly6C to distinguish the different subpopulations of monocytes. Mice injected with MDP had a strong increase in the number of patrolling monocytes associated with a decrease in inflammatory monocytes. Indeed, patrolling monocyte numbers increased by 2-fold (** *p* = 0.0034) when the number of classical monocytes was divided by 2 (** *p* = 0.0097) (Figure 2A). Since MDP binds NOD2, we wanted to confirm that NOD2 is involved in the switch. We used NOD2^−/−^ mice at 3 months of age (*n* = 4) and we injected them with MDP following the same protocol as the WT. Interestingly, after quantification, the population of patrolling monocytes were identical before and after the injections, meaning no switch was visible using flow cytometry (Figure 2B) and showing that MDP triggers the conversion in an NOD2-dependent manner; however, it is interesting that an increase in the population of intermediate monocytes was observed. 

To further characterize the effect of MDP on monocytes, we aimed to evaluate whether the switch was maintained for at least one week since we injected APP mice once a week for 3 months. We used WT at 3 months of age (*n* = 4). After priming the switch, we sampled blood every day for 5 days and analyzed the samples using flow cytometry. The results showed that the switch was maintained at a steady level for at least 5 days (Figure 2C). Then, we evaluated whether the action of MDP was dose dependent. We also observed the effect in vivo, where 5 and 10 mg/kg had a similar effect on the switch; however, for a dose of 1 mg/kg or below, the switch decreased (Figure 2D). Interestingly, we used HEK-NOD2 cells; this cell line is particularly useful since we can monitor the level of NOD2 activation. We incubated the cells with MDP for 6 h with different concentrations at 10, 8, 5, 3, 1, 0.5, and 0.1 µg/mL and using optical density we calculated the level of activation. The quantification showed that the MDP effect is indeed dose dependent since the best induction of NOD2 was found at 8 µg/mL (Figure 2E). 

All these results suggest that MDP via NOD2 is effective in inducing the conversion of monocytes since the increase in the patrolling monocyte number was associated with the drastic diminution in classic monocytes, and the priming followed by one injection every week was enough to maintain the switch all along the protocol with APP mice. 

### 3.2. MDP Has a Powerful Impact on the Short-Term Memory of APP Mice

The APP mouse is a great model of AD for studying the amyloidopathy since mice display cognitive impairment at 6 months of age. The principal manifestation of this decline is a short-term memory loss and CAA [4,16,23,24]. We used the novel object recognition task (NOR) to evaluate the short-term memory at 3 months of age before mice displayed symptoms and at 6 months of age (Figure 1). The test consisted of two distinct phases, namely the familiarization phase and the test phase, and both phases were 1 h apart from each other. We used 8 mice per group, and animals that did not explore each object for at least 10 s during the familiarization phase were removed from the group. For quantification, we used the discrimination ratio (0–1): above 0.5, the mouse explored the novel more during the test phase; and below 0.5 or equal, the mouse spent the same time exploring both objects or the old object more. As expected, WT mice spent more time exploring the novel object at 6 months of age unlike APP mice, which explored both objects for the same amount of time, suggesting an impairment of the short-term memory (♂** *p* = 0.0011; ♀** *p* = 0.0025). Interestingly, MDP-treated mice performed the same as WT mice (Figure 3), suggesting that triggering of patrolling monocytes can delay the cognitive decline observed in the APP mouse at 6 months of age in females and males.

### 3.3. Long-Term Administration of MDP Has a Different Effect on Amyloid Regarding the Biological Sex

Amyloid has an important role in AD pathology since it is thought to be the main cause of the disease. To evaluate whether the MDP has an impact on the amyloid load, we quantified different parameters such as the volume and the number of amyloid plaques, the amount of blood vessels positive for amyloid, and the concentration of Aβ_42_ and Aβ_40_ in the brain.

We used the hippocampus and the cortex as references to count the number of plaques and evaluate their volume. This quantification was carried out using SRC, a stereology software that provides an unbiased count. 

Figure 4A shows a decrease in the number of plaques in the hippocampus and cortex of males treated with MDP at 6 months of age compared to the control littermates (*hippocampus* ♂* *p* = 0.0247; *cortex* ♂ **** p* = 0.0004). On the other hand, females from both groups showed an equivalent number of plaques in the cortex, but we observed a drastic increase in the plaque number in the hippocampus of MDP-treated females (hippocampus ♀ * *p* = 0.0247) (Appendix A). The number of plaques is important, but we also needed to evaluate the volume of those plaques and we found that for either males or females treated with MDP, the volume of plaque remain unchanged compared to the control littermates (Figure 4B), suggesting that MDP do not influence the volume of plaques.

As the CAA is, most of the time, associated with AD, we co-stained blood vessels using CD31 and 6E10 to evaluate the CAA in the brain. Results showed that MDP-treated males had a tremendous increase in blood vessels positive for amyloid (♂* *p* = 0.0323) compared to the non-treated group, but both female groups had an equivalent number of these positive blood vessels (Figure 4C, Appendix A). To further study the CAA, we used ELISA on brain lysates and the quantification showed that males treated with MDP had a higher concentration of Aβ_40_ (**p* = 0.0117) and a lower concentration of Aβ_42_ (♂**p* = 0.0396), leading to a decreased Aβ_42_/Aβ_40_ ratio (♂****p* = 0.0004) (Figure 4D,E), suggesting that amyloid is primarily in blood vessels. However, females of both groups had a light diminution in the concentration of Aß_40_
*(*♀ *p* = 0.0745), whereas MDP-treated females had a slight diminution in Aβ_42_ (♀**p* = 0.0130), but the Aβ_42_/Aβ_40_ ratio was not different between groups (♀ *p* = 0.1408) (Figure 4D,E). This suggests that amyloid is in equilibrium between the brain and the periphery, which is corroborated by the expression of ABCB1 and LRP1, which are the two main molecules able to bind amyloid and transport it from the brain to blood vessels. Indeed, in males treated with MDP, LRP1 and ABCB1 were both overexpressed compared to their controls (ABCB1 ♂**p* = 0.0222; LRP1 ♂**p* = 0.0217) (Figure 4F,G). On the other hand, in females treated with MDP, LRP1 and ABCB1 were both downregulated (ABCB1 ♀**p* = 0.0491; LRP1 ♀**p* = 0.0201) (Figure 4F,G), suggesting an impairment in the clearance via the BBB in MDP-administered females.

### 3.4. MDP Influences the Behavior of Microglia in a Sex-Dependent Manner

Microglia play an important role in defending the brain against aggressor and toxic molecules. To study the impact of MDP on microglia, we quantified the total number of microglia and the number of microglia per plaque using the same stereology software and we used Western blotting to evaluate the phagocytic response of microglia. 

In the male groups, the total number of microglia was statistically identical in the cortex and hippocampus (Figure 5A), although MDP-treated females had a reduced number of microglia in the cortex compared to control littermates (cortex ♀**p* = 0.0227) (Figure 5A). Microglia, in addition to phagocyting amyloid, help to prevent the spread of amyloid by surrounding plaques, leading to the formation of a barrier. In males treated with MDP, we found a reduced number of microglia around plaques in the cortex and hippocampus (cortex ♂****p* = 0.0002; hippocampus ♂**p* = 0.0128) (Figure 5B). MDP-treated females had a higher number of microglia per plaque in the cortex, but not in the hippocampus, compared to the controls (cortex ♀**p* = 0.0227; hippocampus ♀ *p* = 0.8340) (Figure 5B). 

One of the major contributors of the phagocytose and microglial behavior is TREM2. The latter can directly or indirectly bind amyloid to either induce the phagocytose or prevent the spread of amyloid. In both MDP-treated groups, TREM2 expression was tremendously increased compared to the respective controls (♂**p* = 0.0171; ♀**p* = 0.0482) (Figure 5C), but it seemed to have a different effect on male and female microglia. Indeed, we quantified the expression of two molecules involved in the phagocytose pathway, namely LAMP2 and CD68. We found that these molecules were overexpressed in MDP-treated males compared to APP controls, but female MDP and controls had the same level of CD68 and LAMP2 expression (*CD68*: (♂****p* = 0.0003; ♀ *p* = 0.4210); *LAMP2*: (♂**p* = 0.0269; ♀ *p* = 0.6854)) (Figure 5D,E). Moreover, we also quantified arginase-1 (Arg-1), a molecule involved in the regulation of inflammation, and we found that Arg-1 was overexpressed in both males and females treated with MDP (*Arginase-1*: ♂**p* = 0.0185; ♀**p =* 0.0496) (Figure 5F).

These results suggest that MDP has a different effect on microglia, leading to two different behaviors in males and females. 

### 3.5. MDP Prevents BBB Degradation in Males and Females with Conservation of Pre- and Postsynaptic Factor

The integrity of the BBB is crucial for the homeostasis of the brain and the BBB is disrupted in AD, leading to a detrimental environment for the brain and allowing the infiltration of inflammatory monocytes. We quantified the expression of some markers of the BBB by Western blot, namely Claudine 5 and ZO-1, to evaluate whether an increase in the patrolling monocyte population can help to maintain the BBB. We found that both Claudine 5 and ZO-1 were expressed at higher levels in males and females treated with MDP compared to their respective littermate controls (*Claudine-5*: (♂**p =* 0.0178; ♀**p* = 0.0366); *ZO-1* (♂**p* = 0.0161; ♀****p* = 0.0002)) (Figure 6A,B).

Then, we evaluated whether MDP had an impact on neurons and synapses. We quantified the expression of PSD-95, SMI-312, and BDNF. These molecules are informative of the health of the brain since PSD-95 provides information synapses, SMI-312 neurites, and, finally, BDNF is a key molecule involved in neuronal plasticity and learning. We found that PSD-95 expression was increased in males treated with MDP and in females treated with MDP compared to their respective controls (PSD-95: ♂*****p* < 0.00001); ♀**p* = 0.0314) (Figure 6C). This increase was also associated with an augmentation in SMI-312 and BDNF in both sexes (*SMI-312*: ♂**p* = 0.0104); ♀**p* = 0.0457; *BDNF*: ♂**p* = 0.0414); ♀**p* = 0.0107) (Figure 6D,E), meaning a lower number of dystrophic neurites and better support from microglia to neurons. 

Altogether, these results suggest that MDP, via patrolling monocytes, can protect the BBB by maintaining the tight junctions between cells at the BBB level and can also protect neurons and synapses by maintaining the expression of PSD-95, SMI-312, and BDNF. 

## 4. Discussion

In this study, we aimed to study the role of MDP in patrolling monocytes and microglia in AD. For this purpose, we used the APP_swe/PS1_ mouse model, and we injected MDP for 3 months between 3 and 6 months of age. AD is a multifactorial disease in which the innate immunity, such as microglia and patrolling monocytes, plays a major role. MDP is a component of the bacterial cell wall; the latter binds to NOD2 and induces a switch in the monocyte population from classic to patrolling [18]. A recent study used an NOD2 knockout mouse strain to study the role of NOD2 in the switch. Lessard and colleagues found no difference in the population of patrolling monocytes between WT mice injected with saline and NOD2^−/−^ mice injected with MDP [18], although the population of intermediate monocytes increased. The exact mechanism underlying the conversion is not yet understood, but we can hypothesize that classical monocytes are blocked during the conversion at an intermediate stage in the context of NOD2 gene deletion. Moreover, this hypothesis is sustained by the expression of markers at the surface of this population, which are between classical and patrolling monocytes [25].

It is worth mentioning that patrolling monocytes have been poorly studied in the context of AD and CAA. The only study to our knowledge studying the role of MDP in male APP mice was carried out by Maleki and colleagues, which showed that MDP was effective in maintaining the switch over three months with one injection per week. Moreover, studies have suggested that such a monocyte population is in contact with the amyloid in blood vessels and, therefore, can help to clear amyloid deposits and can synthesize growth factors and anti-inflammatory factors and trigger the sink effect [15,26,27]. The sink effect is a theory based on the osmotic rules; the idea is that amyloid is in equilibrium between the brain and blood vessels and removing amyloid from the periphery should lead to an efflux of amyloid from the brain [28,29]. To further study the role of MDP, we also assessed its role in both sexes, as we know that AD has a different impact on immune cells in females and males [2,4,30,31]. 

Mice were treated before the manifestation of cognitive decline at 3 months of age with 1 injection per day for 3 consecutive days to induce the switch and then once a week. APP mice bear two different human mutations, which induce overproduction of the amyloid load in both the brain and BBB. At 3 months of age, APP mice are at the early stage of the disease. By starting the treatment at this time, we aimed to target patients with mild cognitive impairment and study whether MDP can effectively delay cognitive decline. For the first time, we demonstrated that MDP can delay shot-term memory loss at 6 months of age (Figure 3) in both sexes. NOR is the gold standard test [23,32] for assessing cognitive decline and APP mice were able to differentiate between the novel object and the familiar object in response to MDP. This result suggests that triggering patrolling monocytes in APP mice has a beneficial effect on cognition, which is not reflected by the same outcomes regarding the count and the volume of amyloid plaques in both sexes. Indeed, females and males have a different profile regarding the amyloid load and microglia (Figure 4 and Figure 5). Males treated with MDP displayed a lower number of plaques associated with an increased CAA, suggesting that the main mechanism of elimination is from the periphery via the sink effect and patrolling monocytes. These cells act as blood vessel housekeepers since they are mobilized by inflammatory cues from amyloid micro-aggregates in the vasculature and they crawl toward these deposits to remove them [33]. ABCB1 and LRP1 are two key transporters in the BBB that bring amyloid peptides out of the brain across endothelial cells, a phenomenon called the sink effect. MDP seems to be quite efficient in stimulating such a mechanism, especially in male mice.

Many studies have used different approaches to stimulate the sink effect. Although some immunotherapies were able to produce beneficial changes in the amyloid load, these changes were, most of the time, associated with micro bleeding, micro hemorrhage, and BBB breakdown, which can worsen the disease [34,35,36]. Here, we demonstrated that MDP is safe for a 3-month period since mice did not exhibit health issues or show adverse effects. Interestingly, the microglia of males treated with MDP overexpressed phagocytic markers such as TREM2, CD68, and LAMP2, suggesting that microglia are activated and can effectively remove amyloid in the brain as well. Indeed, TREM2 is a key component of the microglial response and activation and is essential for their metabolism to detect and respond to neurodegenerative cues and it is involved in numerous pathways such as cell survival and homeostasis [23,37]. Studies have also shown that a mutation or a decrease in TREM2 expression is associated with a downregulation of CD68 [38,39], whereas TREM2 overexpression triggers Arg-1 expression [40].

The expression of Arg-1 was increased in the treated group compared to control littermates and a recent published article showed a link between Arg-1 and Aβ restriction [41]. It is possible that MDP increases Arg-1 production by microglia to restrict Aβ accumulation, which seemed to be the case in the brain of female mice. We, therefore, suggest that MDP has the ability to modulate the behavior of microglial cells and mitigate their inflammatory properties since Arg-1 is described as an anti-inflammatory factor [41,42,43]. However, studies from the past years demonstrated that the description of the microglial phenotype is not as simple and can often be misleading since microglia can quickly alternate between inflammatory and anti-inflammatory phenotypes [44,45].

AD affects males and females differently and the results for this study showed that MDP improves animal conditions with potential different mechanisms. Indeed, in females, the effect seems restricted to the brain compared to the sink effect detected in males (Figure 4, Figure 5 and Figure 6). Microglia in non-ovariectomized mice are more protective than in older counterparts, suggesting a role of sexual hormones in these effects, at least in female APP mice [46]. A similar pattern has been observed for TREM2 and Arg-1 expression but not for CD68 and LAMP2. Consequently, it is possible to suggest that microglia act as a barrier to avoid the spread of amyloid instead of phagocytosis in females treated with MDP [47,48]. Although we highlight here some differences between males and females, little is known about the behavior of microglia between sexes, especially in the context of AD.

The BBB is crucial to maintaining the homeostasis of the brain and we observed a degradation of the BBB during the course of AD that can potentially accelerate the disease [49,50]. However, MDP was able to improve such degradation in both treated groups and this may contribute to delaying their cognitive decline. 

Altogether, these results shed new light on the beneficial effect of MDP and the stimulation of NOD2 in AD. For the first time, we demonstrated that chronic injection of MDP can improve cognition in both sexes. In stimulating patrolling monocytes and microglia, MDP was able to trigger the sink effect and the phagocytosis pathways, which was associated with a reduced number and volume of amyloid plaques in male animals. The effect of MDP seems to be more restricted to the brain parenchyma, with a better protective role of microglia surrounding amyloid plaques in the female group.

Further studies are required to fully understand the mechanisms of MDP in males and females, including genomic studies; however, clearly, NOD2 ligands are quite good candidates for future therapy in both sexes. 

## Figures and Tables

**Figure 1 cells-11-02241-f001:**
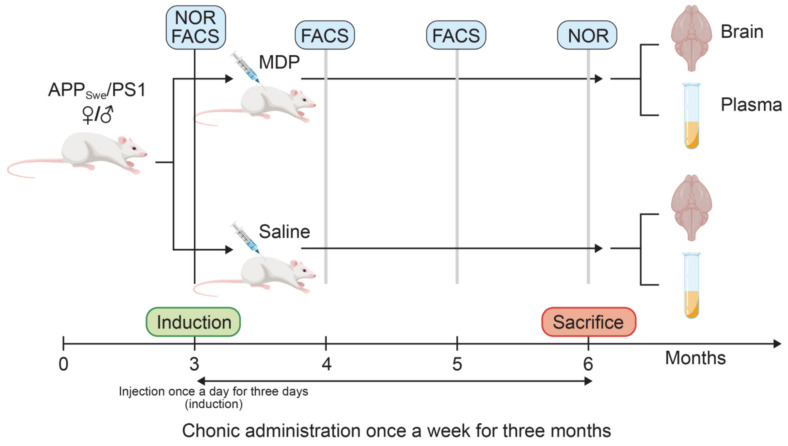
Timeline of the experimental design and methods following MDP (10 mg/kg) or saline (0.9%) administration in male or female wild-type and APP_Swe_/PS1 treated or not with MDP from 3 to 6 months old. Mice were injected once a day for three days (induction) followed by one injection a week for three months. At 6 months, the mice were sacrificed, and their brains and the plasma were extracted for further analysis.

**Figure 2 cells-11-02241-f002:**
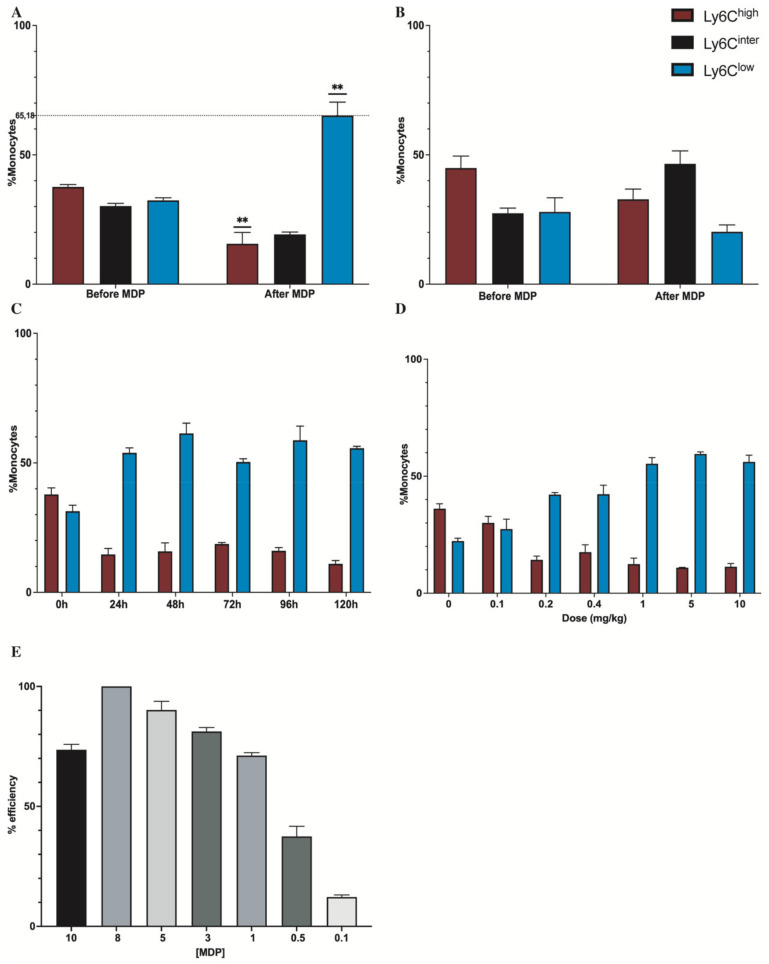
MDP triggers the monocyte conversion in an NOD2-dependent manner. (**A**): Flow cytometry analysis of monocyte subsets in blood on (**A**) WT and (**B**) NOD2^−/−^, before and after administration of an injection of MDP each day on 3 consecutive days at 10 mg/kg. (**C**): Flow cytometry analysis of monocyte subsets in the blood before MDP administration 0 to 120 h post-induction. (**D**): Dose response after induction of MDP at 0, 0.1, 0.2, 0.4, 1, 5, and 10 mg/kg on WT mice. (**E**): Results of MDP are dose dependent on HEK-Blue™-hNOD2 cells in contact for 6 h. Data are mean ± SEM (*n* = 4 animals/group). ** *p* < 0.01 compared to the monocyte subset population before induction.

**Figure 3 cells-11-02241-f003:**
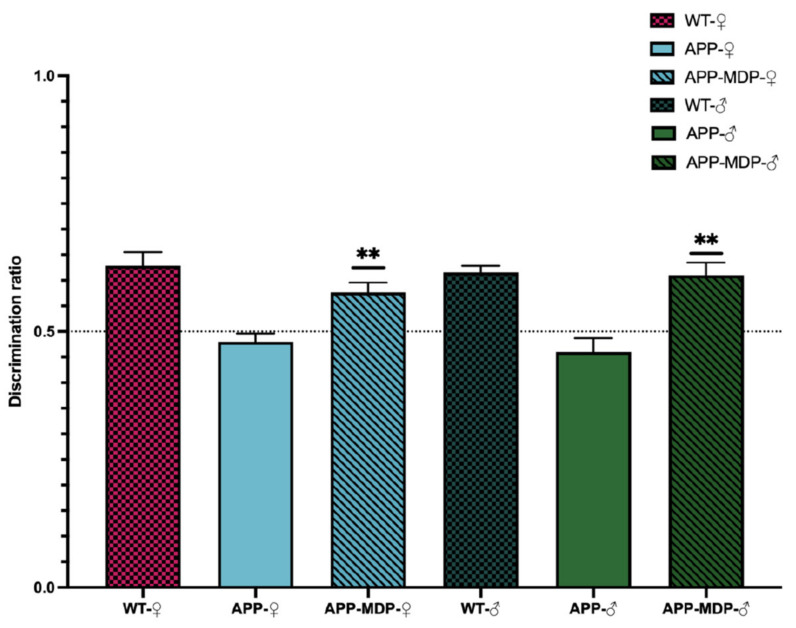
MDP has a powerful impact on the short-term memory of APP mice. Novel object recognition tasks with 2-time-point comparison of male or female WT and APP_Swe_/PS1 from 3 to 6 months. Discrimination differs from 0.5. Data are mean ± SEM (*n* = 8 animals/group). ** *p* < 0.01 compared to sex-matched APP_Swe_/PS1 controls.

**Figure 4 cells-11-02241-f004:**
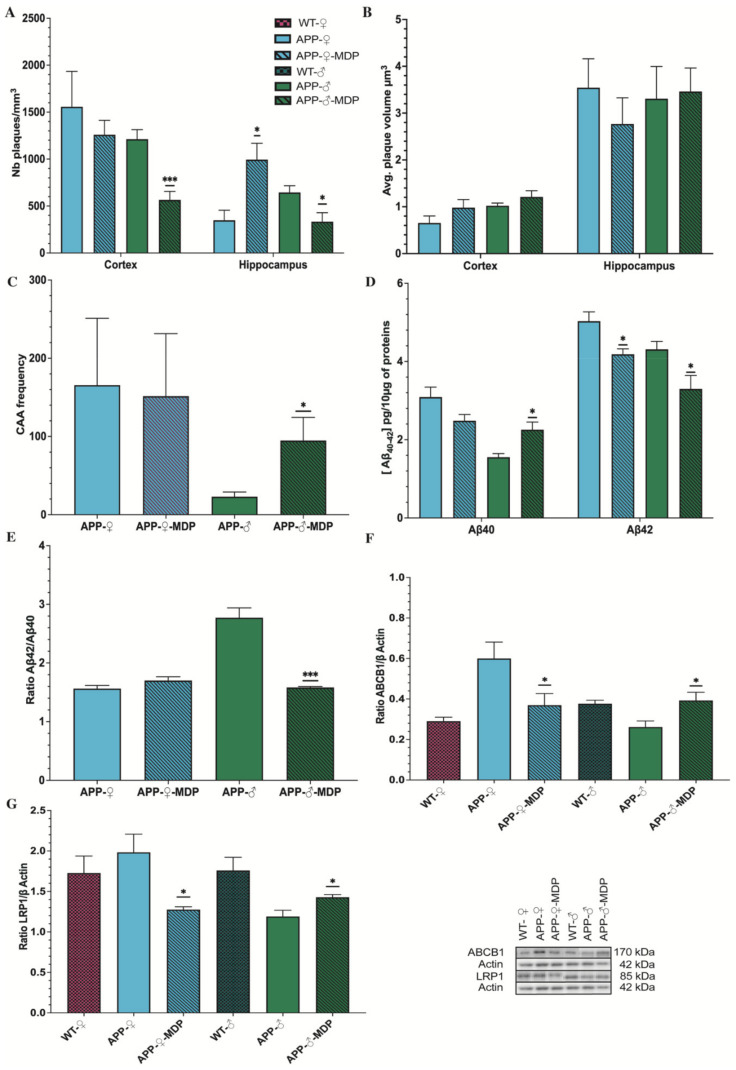
The long-term administration of MDP has a different effect on amyloid regarding the sex. Although, MDP induces an increase in the plaque number in the female APP_Swe_/PS1 treatment with MDP group. (**A**): Unbiased stereological analysis of the 6E10^+^ plaque number of the male and female APP_Swe_/PS1 treatment or not with MDP groups in the hippocampus and cortex. Analysis showed a diminution in the plaque number in the male APP_Swe_/PS1 treatment group in both areas and an augmentation in the Aβ plaque numbers in the female APP_Swe_/PS1 treatment group in the hippocampus compared to sex-matched APP_Swe_/PS1 controls. (**B**): Average volume per 6E10^+^ plaque in µm^3^ of the male and female APP_Swe_/PS1 treatment or not groups with MDP in the hippocampus and cortex. Analysis showed no difference in both sexes and area compared to sex-matched APP_Swe_/PS1 controls. (**C**): The 6E10+ blood vessel frequency of male and female APP_Swe_/PS1 treatment or not with MDP groups. Analysis showed an augmentation in the male APP_Swe_/PS1 treatment group but no difference in the female APP_Swe_/PS1 treatment groups compared to sex-matched APP_Swe_/PS1 controls. (**D**): Elisa Aβ40 and Aβ42 of the male and female APP_Swe_/PS1 treatment or not with MDP groups. Analysis showed an increase in Aβ40 in the male APP_Swe_/PS1 treatment group but a decrease in the Aβ42 in APP_Swe_/PS1 treatment group for both sexes compared to sex-matched APP_Swe_/PS1 controls. (**E**): Ratio of Aβ42/Aβ40 in the male and female APP_Swe_/PS1 mice treatment or not with MDP groups. Analysis showed a diminution in the male APP_Swe_/PS1 treatment group but no difference in the female APP_Swe_/PS1 treatment group compared to sex-matched APP_Swe_/PS1 controls. (**F**); (**G**): MDP increases the clearance systems through LRP1 and ABCB1 expression in the male group. Western blot analysis of (**F**) LRP1 (**G**) ABCB1 in the male and female wild-type and APP_Swe_/PS1 treatment or not with MDP groups. Analysis showed an augmentation in LRP1 and ABCB1 expression in the male APP_Swe_/PS1 treatment group but a diminution in LRP1 and ABCB1 expression in the female APP_Swe_/PS1 treatment group compared to sex-matched APP_Swe_/PS1 controls. Data are mean ± SEM (*n* = 8 animals/group) * *p* < 0.05, *** *p* < 0.001 compared to sex-matched APP_Swe_/PS1 controls.

**Figure 5 cells-11-02241-f005:**
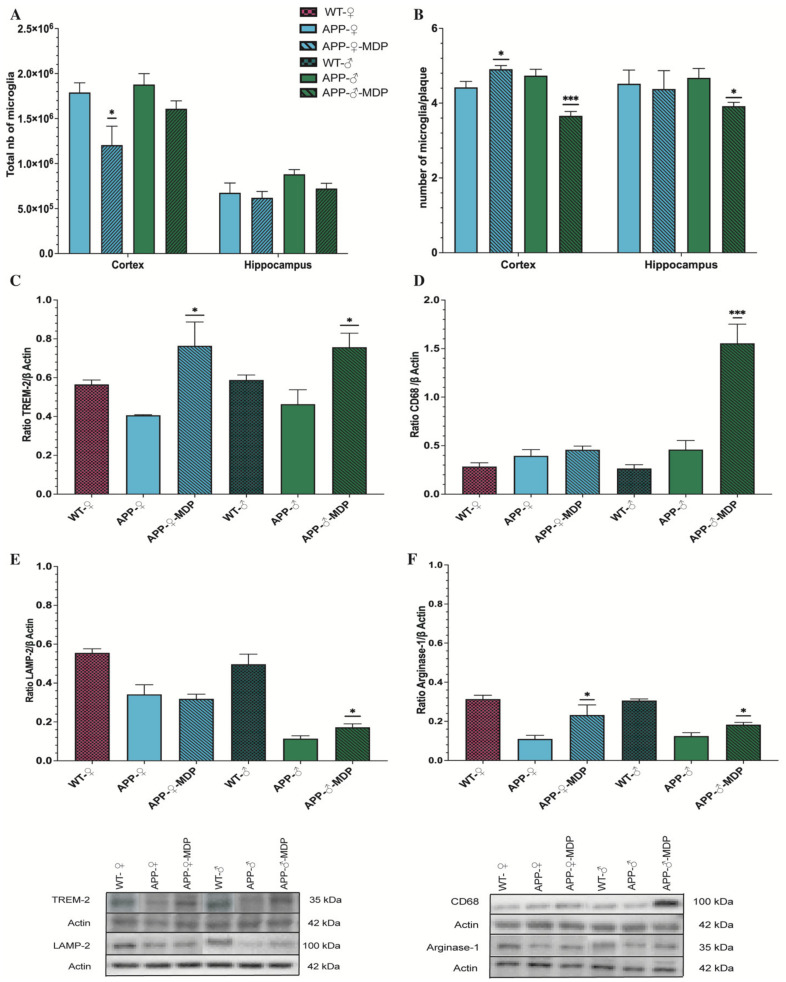
MDP influences the behavior of microglia in a sex-dependent manner. (**A**): Unbiased stereological analysis of the microglia (Iba-1) number in the male and female APP_Swe_/PS1 treatment or not with MDP groups in the hippocampus and cortex. Analysis showed no difference in the male APP_Swe_/PS1 treatment group in both areas; microglia are less present in the female APP_Swe_/PS1 treatment group in the cortex compared to the sex-matched APP_Swe_/PS1 controls. (**B**): Number of microglia per 6E10^+^ plaque of the male and female APP_Swe_/PS1 treatment or not with MDP groups in the hippocampus and cortex. Analysis showed a decrease in the male APP_Swe_/PS1 treatment group in both areas but only an augmentation for the female APP_Swe_/PS1 treatment group in the cortex compared to sex-matched APP_Swe_/PS1 controls. (**C**–**F**): Western blot analysis of (**C**) TREM2, (**D**) CD68, (**E**) LAMP2, and (**F**) Arginase-1 in the male or female wild-type and APP_Swe_/PS1 treatment or not with MDP groups. Data are mean ± SEM (*n* = 8 animals/group). * *p* < 0.05, *** *p* < 0.001, compared to sex-matched APP_Swe_/PS1 controls.

**Figure 6 cells-11-02241-f006:**
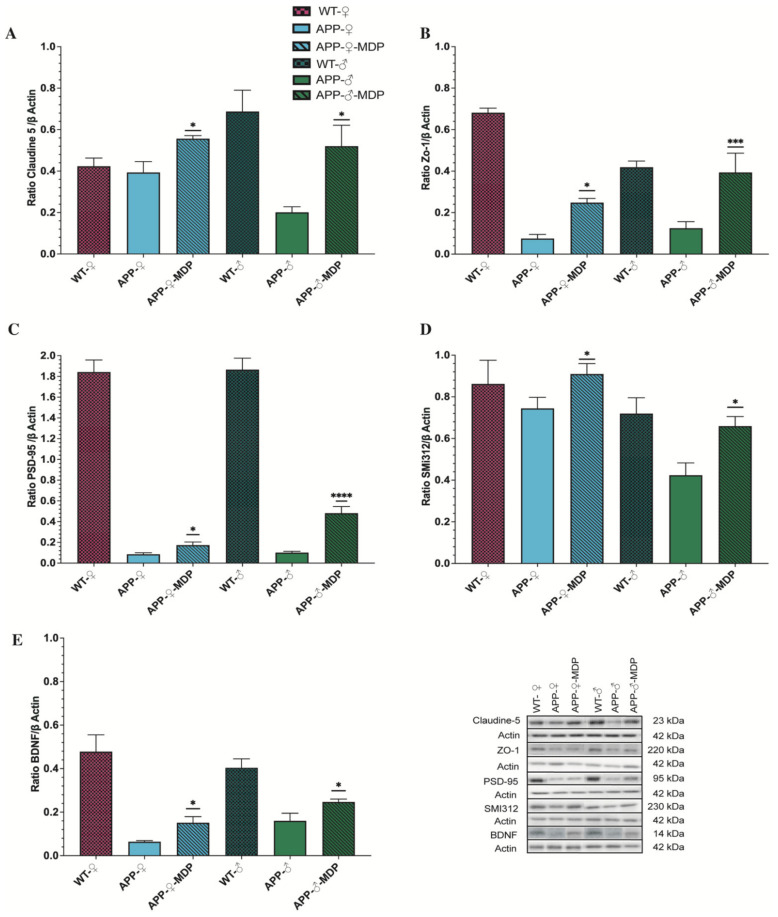
MDP prevents BBB degradation in males and females with conservation of pre- and postsynaptic factor. Western blot analysis of (**A**) Claudin-V, (**B**) ZO-1, (**C**) PSD-95, (**D**) SMI312, and (**E**) BDNF of the male or female wild-type and APP_Swe_/PS1 treatment or not with MDP groups. Data are mean ± SEM (*n* = 8 animals/group). * *p* < 0.05, *** *p* < 0.001, **** *p* < 0.0001 compared to sex-matched APP_Swe_/PS1 controls.

**Table 1 cells-11-02241-t001:** List of antibodies used for the Western blot and immunostaining analyses and all related information, including the name of the company, molecular weight, species, primary and secondary antibody dilution, and targets.

Antibody	Company	Molecular Weight	Species	Cat#	Concentration	Secondary Antibody	Cat#	Concentration
6 E 10	Biolegend	NA	Mouse	SIG-39320	1/1000	IgG anti-Mouse Cy3	115-165-003	1/1000
ABCB1	Novus biological	170 kDa	Rabbit	NBP2-67667	1/1000	IgG anti-Rabbit peroxidase	111-035-144	1/2000
Actin	Sigma-Aldrich	42 kDa	Mouse	A5316	1/5000	IgG anti-Mouse peroxidase	115-035-003	1/5000
Arginase-1	Invitrogen	35 kDa	Rabbit	702730	1/1000	IgG anti-Rabbit peroxidase	111-035-144	1/2000
BDNF	Millipore	14–17 kDa	Rabbit	AB-1534	1/1000	IgG anti-Rabbit peroxidase	111-035-144	1/2000
CD68	Serotec	100 kDa	Rat	MCA155	1/1000	IgG anti-Rat peroxidase	112-035-003	1/800
CD31	BD Bioscience	NA	Rat	550274	1/1000	IgG anti-Rat AF488	A11006	1/1500
Claudine-5	Abcam	23 kDa	Rabbit	AB15106	1/1000	IgG anti Rabbit peroxidase	111-035-144	1/2000
Collagen IV	EMD	NA	Goat	AB769	1/1000	IgG-anti-Goat AF488	A21467	1/1500
Iba-1	Wako	NA	Rabbit	019-19741	1/1000	IgG anti-Rabbit A488	A11008	1/1000
LAMP-2	Thermo Fisher	100 kDa	Rabbit	PA1-655	1/1000	IgG anti-Rabbit peroxidase	111-035-144	1/2000
LRP1	Abcam	85 kDa	Rabbit	AB-92544	1/1000	IgG anti-Rabbit peroxidase	111-035-144	1/2000
PSD95	Millipore	95 kDa	Mouse	MAB-1596	1/1000	IgG anti-Mouse peroxidase	115-035-003	1/1000
SMI-312	Covance	230 kDa	Mouse	SMI-32	1/1000	IgG anti-Mouse peroxidase	115-035-003	1/2000
Trem-2	R&D system	35 kDa	Sheep	AF-1729	1/1000	IgG anti-Sheep peroxidase	313-295-003	1/2000
ZO-1	Thermo Fisher	220 kDa	Rabbit	61-7300	1/1000	IgG anti-Rabbit peroxidase	111-035-144	1/2000

## Data Availability

Data will be made available by the corresponding author on reasonable request.

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
