# Peer review of "Muramyl Dipeptide Administration Delays Alzheimer’s Disease Physiopathology via NOD2 Receptors"

_cells, 2022, doi:10.3390/cells11142241_

Round 1

Reviewer 1 Report

The authors evaluated the effect of the NOD2 receptor ligand muramyl dipeptide (MDP) on the modulation on the innate immune cells from male and female APP/PS1 mice. It was observed that MDP improve cognitive function and the BBB in both male and female mice but through distinct mechanisms.

There observations are interesting but clarifications and improvements are required:

Mice experimentation:

- the reference of the protocol approved by the Ethics Committee must be included.

- the number of mice (female and male) per group must be indicated.

List of antibodies:

- the antibody catalogue code should be included in the list

- the full information in relation to the secondary antibodies is required (and not only the dilution)

Figure 2 can be included as supplemental figure.

Statistical analysis - did the authors tested their data for normality?

Results:

Figure 4 - The authors indicate that blood samples were analysed by flow cytometry using CD11b, CD45, Ly6C antibodies. However, only Ly6C results are shown. Is there any reason for that?

The authors indicate that no alterations in monocytes subsets were observed in NOD2-/- mice treated with MDP. However, some alterations are observed and should be discussed.

These results were obtained from MDP treated WT mice. Do the authors expect the same behavior in diseased mice (APP/PS1)? Please discuss.

Figure 5 - In some graphs is difficult to identify the mice group studied. Authors must add a clear legend to identify the bars (similar to figure 6).

In some graphs there is a clear difference between biological sex per se (including data from other figures). This should be clearly acknowledged in results section and discussed.

In the legend of figure, the information regarding the number of mice used in the determination of the parameters is not clear.

An important aspect that should be evaluated is the phenotype of microglia in untreated male/female mice vs treated mice (microglial pro- and anti-inflammatory phenotype).

(Neuro)inflammation is also characterized by astrocytosis. Did the authors have any idea if MDP also decreases astrocytosis.

A major issue with this study is that the mechanistics underlying the benefits of MDP, particularly in females, is poorly explored.

Author Response

Reviewer 1 :

Thank you for your comments, we appreciated the time spent on our MS

  • Comment 1 “Mice experimentation:

- the reference of the protocol approved by the Ethics Committee must be included.

- the number of mice (female and male) per group must be indicated.”

Response: We included the reference within the text, and we mentioned the number of mice in this section. See the response added to the MS (red).

Comment 2 “List of antibodies:

- the antibody catalogue code should be included in the list

- the full information in relation to the secondary antibodies is required (and not only the dilution)”

Response: Please find the modification in the table (red)

Comment 3 “Figure 2 can be included as supplemental figure”

Response: Please find the supplementary figure modified

Comment 4 “Statistical analysis - did the authors tested their data for normality?”

Response: please find the modification in red within the text

Comment 5 “ Figure 4 - The authors indicate that blood samples were analysed by flow cytometry using CD11b, CD45, Ly6C antibodies. However, only Ly6C results are shown. Is there any reason for that?”

Response: We used CD11b, CD45 to select specifically monocytes and then we analyzed the switch with Ly6C which is a reliable marker to study monocytes. Please find the explanation in the text (red)

Comment 6 “The authors indicate that no alterations in monocytes subsets were observed in NOD2-/- mice treated with MDP. However, some alterations are observed and should be discussed.”

Response: Sentences have been added to the results and discussion section. However, little is known about the mechanism of switch and the different population of monocytes in NOD2 mice. (red)

Comment 7 “These results were obtained from MDP treated WT mice. Do the authors expect the same behavior in diseased mice (APP/PS1)? Please discuss.”

Response: The switch has been assessed in APP at 4 and 5 months of age (figure 1/submitted version) and previously described in male (Maleki et al., 2021). Please find the response in the text (red).

Comment 8 “Figure 5 - In some graphs is difficult to identify the mice group studied. Authors must add a clear legend to identify the bars (similar to figure 6).”

Response: We modified the figures as suggested

Comment 9 “ In some graphs there is a clear difference between biological sex per se (including data from other figures). This should be clearly acknowledged in results section and discussed.”

Response: This is a well-known fact in biology and we have tried to address when appropriate.

Comment 10 “In the legend of figure, the information regarding the number of mice used in the determination of the parameters is not clear.”

Response: the legends have been updated

Comment 11: “An important aspect that should be evaluated is the phenotype of microglia in untreated male/female mice vs treated mice (microglial pro- and anti-inflammatory phenotype).”

Response:  When we wrote the discussion as well as result sections, we were really careful using the term phenotype since more and more evidence show that M1/M2 or pro-anti inflammatory phenotypes is not a clear as it was the last 2-5 years. We mentioned that microglia in males and females overexpress Arg-1 which is an anti-inflammatory factor. TREM2 could also play the role of anti-inflammatory factor but we do not have enough evidence to clearly state that microglia are in one or another state of activation.

Comment 12: “(Neuro)inflammation is also characterized by astrocytosis. Did the authors have any idea if MDP also decreases astrocytosis.”

Response: This is an excellent question, for this particular study, we focused on microglia and monocytes.

Comment 13: “A major issue with this study is that the mechanistics underlying the benefits of MDP, particularly in females, is poorly explored”

Response: We totally agree with this comment. That is the major flaw of this study, with this article we aimed to highlight the differences between males and females. AD is quite complex, adding sex differences is even harder. This study is the first stone, the proof of concept, we need more evidence from the genomic and proteomic to discover what can explain those differences.

Reviewer 2 Report

The manuscript by Piec etc al. reports that treatment with muramyl dipeptide improves cognitive function in alzheimers mice. Overall the study is interesting and performed well. The authors report clear improvement in short term memory of both male and female mice and show that in both sexes the treatment reduces BBB breakdown. The results are interesting although the effect on cognitive function has been seen earlier in male mice and the changes seen in a-beta levels and microglia in male mice may not explain the improved function as these are not seen in female mice. 

The effect of the treatment on amyloid beta and microglia are somewhat  confusing as the effect is different between the biological sexes and in many cases it seems like even though the MDP treatment seems to have even opposite effects on male and female mice, the sex differences seen in untreated mice are abolished in the treated mice. Understanding the significance of these results would benefit  from including the wt mice data in the figures, this is now only shown for few parameters. 

Also for the western and immunohistochemistry data, the actual stainings should bee included at least in supplementary. Now only the quantification of the data is shown, and it is difficult to assess the data when the quality of the stainings is not shown.

Minor comments:

-The flow cytometry chapter more resembles a protocol than method description and could be modified

-Figure 4 comes before figure 3

-Figure 4 c and d, why only two of the three monocyte types are shown?

-Fig 4e The log scale makes it hard to understand the figure

-Figure 5  is the data from 5-8 animals, or from 6-7 animals per group, both are mentioned in the legend.

Author Response

Reviewer 2:

Thank you for your comments, we appreciated the time spent on our MS

Comment 1: “Understanding the significance of these results would benefit from including the wt mice data in the figures, this is now only shown for few parameters.”

Response: We described and highlighted the major differences between males and females. AD is complex, adding sex-differences is even harder. However, we can as we did in the discussion hypothesized on what could be the cause/meaning of the results.

We added all WT data in the figures. Except for the figure of stereology since we did not quantify the amyloid in WT for obvious reason.

Comment 2: “Also for the western and immunohistochemistry data, the actual stainings should bee included at least in supplementary. Now only the quantification of the data is shown, and it is difficult to assess the data when the quality of the stainings is not shown.”

Response: We modified this according to your suggestions.

Comment 3: “-The flow cytometry chapter more resembles a protocol than method description and could be modified”

Response: Please find the modification in red

Comment 4: “Figure 4 comes before figure 3”

Response: we modified that

Comment 5: “Figure 4 c and d, why only two of the three monocyte types are shown?”

Response:  We present the 3 types of monocytes only in those figures to show that inflammatory monocytes switch to patrolling monocytes. Little is known about intermediate monocytes we did not want to speculate to much on this population. However, we added in the discussion few lines about the intermemediate monocytes (red)

Comment 6 “Fig 4e The log scale makes it hard to understand the figure”

Response: we have changed the scale

Comment 7: “Figure 5  is the data from 5-8 animals, or from 6-7 animals per group, both are mentioned in the legend.”

Response: We did update the legend